# Exploring the Relationship between Early Leaving of Education and Training and Mental Health among Youth in Spain

Laura M. Guerrero-Puerta [1,2,3,4] and Miguel A. Guerrero [5,*]

1   Department of Social Psychology and Education, University Pablo de Olavide, 41013 Seville, Spain;
    laura.guerrero.puerta@gmail.com
2   Department of Didactics, International University of La Rioja, 26006 Logroño, Spain
3   Department of Education and Social Psychology, University Isabel 1 de Castilla, 09003 Burgos, Spain
4   Reseach Group HUM-308, University of Granada, 52005 Granada, Spain
5   Independent Researcher, 29680 Málaga, Spain
*   Correspondence: migupu97@gmail.com

**Abstract:** This study aimed to explore the relationship between Early Leaving Education and Training and mental health perceived by young Spanish school leavers, as well as develop mid-range theories to better understand this relationship. The study uses a grounded theory approach; specifically, Charmaz's constructivist approach and its informed version have guided the study. Through qualitative interviews with individuals who had left school prematurely, the results of this study showed a bidirectional relationship between ELET and mental health, where the detriment in the mental health of young people who leave school early should be understood as both a cause and an effect of the process of ELET. In addition, the findings suggest that certain socio-economic and educational factors, such as bullying, academic stress, self-responsibilization of failure, and labels such as "NEET" can contribute to a decrease in mental health. Overall, this study has provided new insights into the ELET-mental health relationship, contributing to the development of mid-range theories that can inform future research and interventions to minimize these problems.

**Keywords:** early leaving education and training; mental health; youth; bullying; social isolation; self-esteem; labor aspirations; academic stressors; hyper-responsibilization; NEET

## 1. Introduction

The literature has pointed to the fact that mental health[1] is a critical factor in educational success [2,3]. On the other hand, literature shows that Early Leaving from Education and Training (ELET)[2] has become an increasingly worrying problem in the European Union [5–7], particularly in Spain, where ELET rates have been high for more than ten years and are seen as a hindrance to economic growth and social cohesion both in the country and in the Union [8]. This article examines the overlap between these two topics: exploring the potential causes of this issue and its effects on youth.

An analysis of existing literature [9–14] revealed a connection between mental health issues and ELET; however, limited qualitative research methods have been used to explore this relationship in depth, making it difficult to accurately comprehend underlying factors or draw reliable conclusions about students' perceptions without giving them the opportunity to explain themselves or "have a voice" [15]. In addition to this, the European Union has recognized the connection between mental health, well-being, and education. It has supported and encouraged research and interventions to explore the relationship between mental health and school life, particularly in terms of educational attachment, school failure, and ELET [16,17]. However, it acknowledges that there is still room for improvement in the understanding of risk factors and effective interventions in this area [16]. Thus, this study seeks to provide a deeper understanding of mental health among Spanish youth that suffered from ELET by utilizing grounded theory approaches with qualitative research

methods so as to uncover mid-range theories that could potentially shed some light to better understand this possible association.

Therefore, the main goal of this paper is to analyze the possible relationship between ELET and mental health in young leavers by trying to understand what dynamics influence their decisions as well as how we can protect their psychological well-being[3] after they make such choices. The primary question asked was: what are the perceptions of young leavers on the relationship between mental health and ELET?

We believe that research on the possible relationship between ELET and mental health in young leavers is necessary because it can provide valuable insights into how ELET affects mental health and how best to protect the psychological well-being of young people who have left education and training early. This research can be used to help educators more effectively understand and address the underlying issues behind ELET, which can, in turn, help to reduce the number of young people who leave education and training early. It can also help families and communities recognize and address the needs of those who are vulnerable to leaving early, as well as help identify resources to enhance emotional well-being and reduce the impacts of ELET when it does occur. Additionally, by utilizing qualitative research methods such as grounded theory approaches, this research can uncover deeper insights into the relationship between ELET and mental health that can help inform policy and practice. This can help to ensure that young people are supported and empowered to make the best decisions for their future. Subsequently, the theoretical framework underlying this rationale is explored for a better understanding.

### 1.1. Theoretical Framework

In recent years, there has been an increase in the number of scientific publications, and the national and international media have positioned mental health as an urgent priority for public health [20]. However, many aspects of mental health are not fully understood by youngsters [21]. Therefore, it is essential to discuss the key areas that influence mental health among youth before delving further into this research. Furthermore, we must consider the effects on mental health when young people commit ELET and the academic environment as an influential factor [15]; this is particularly important given our objective in this article. This theoretical framework will delve into these topics to gain a better understanding of mental health among youth and the relation between mental health and ELET.

#### 1.1.1. Mental Health in Young People

Health and youth may have a positive association; however, there are a few issues associated with this, such as drug use, nutrition, and sedentary lifestyles, that suggest otherwise [22,23]. Therefore, it is important to consider topics such as physical activity and sport, the promotion of healthy lifestyles, and mental health among young people [24,25]. Effective interventions are needed in these regards [26].

Studies suggest that even when youth is a period of life when self-perception of mental health is still being developed, there is a lot of confusion surrounding self-awareness of mental health, as adolescents are unfamiliar with the concept of positive mental health and tend to use lay terms to describe conditions, which ultimately can lead to an inaccurate assessment of mental health and hinder appropriate help-seeking [21].

It is concerning that young people are not always capable of assessing the importance of mental health, which can lead to inappropriate help-seeking [21]. Still, more importantly, it is for public institutions to focus on this issue and become key actors for change [26]. The World Health Organization (WHO) has classified various mental health disorders; however, certain conditions, such as "common mental disorder", which refers to the most prevalent conditions, are not included, even when they are terms such as depressive episode, neurotic, stress-related and somatoform disorders [20,27]. This is just an example that shows the need for a better assessment of this issue from public institutions.

### 1.1.2. Effects of the School Environment on Mental Health

A recent literature review [28] has pointed out that, while further research is needed on the topic, there are significant indications to suggest schools can be effective environments for promoting mental health and well-being. This could be attributed to the amount of time children and adolescents spend in school as well as its capacity to provide learning experiences that enable students to build aspects of their identity, establish relationships and develop interpersonal/intrapersonal skills—all of which may help lay a foundation for good mental health in future years [17]. However, it has also been noted that schools can potentially become an environment conducive to the development of mental illness due to phenomena such as bullying, academic stress, or integration issues into the school environment. These problems are known to have long-term implications [29,30].

In response, organizations like UNESCO and the OECD have both recognized the role schools play in promoting adolescent mental health/well-being by creating initiatives such as the "Strategy on Education for Health and Well-Being" (UNESCO) and the "PISA Framework for Analysis of Student Well-Being" (OECD). Govorova et al. [30] outline five main domains within this framework: cognitive well-being; psychological well-being; physical well-being; social well-being, and material well-being, each involving variables relevant to student knowledge/abilities when resolving everyday issues plus perceptions about life/school engagement, etc.

### 1.1.3. Health Effects of ELET

Moreover, focusing on literature, when it comes to mental health, research has identified a possible relationship between ELET and various issues. Liem et al. [31] and Higgins et al. [32] found that those who experienced ELET reported significantly higher percentages of depression at the time of expected graduation than those who graduated from secondary school. Additionally, several studies have detailed that individuals who suffer ELET are more likely to suffer from mood disorders as well as suicidal ideation [9–11]. The World Health Organization also reports an increase in the suicide rate among 15–24-year-olds over the last two decades, making this one of the leading causes of death for this age group globally [33,34]. Nowadays, suicide is the fourth leading cause of death for the population between 15 and 29 years of age [35]. This poses early leavers with an even greater vulnerability.

De Ridder et al. [12], Esch et al. [13], and Hjorth et al. [14] all suggested an association between poor self-rated mental health in adolescence/young adulthood, ELET rates, and reduced work integration; these findings suggest that preventive measures on both individual and societal levels may be necessary to address this issue. Consequently, it is evident that there can be serious implications for individuals' mental health when they prematurely leave schooling. This highlights the need for preventative interventions targeting this population.

In relation to this, it is well established [36–39] that poverty is an essential link between educational disadvantage and health outcomes. Those with higher levels of education tend to earn higher incomes which leads to a better quality of life. In comparison, early school leavers are more likely to be marginalized or socially excluded resulting in lowered levels of well-being or increased mortality rates.

To sum up, the literature points out that there may be a relationship between mental health decline and ELET [12–14]. The physical, mental, and behavioral effects of ELET can have serious implications for young people's well-being [16,17]. It is important to consider topics such as physical activity and sports promotion of healthy lifestyles and mental health among young people in order to prevent negative outcomes associated with ELET [24,25]. Public institutions should focus on this issue by providing educational/formative processes that target the academic sphere as well as emotional education initiatives that address the needs of those who suffer ELET [16]. The literature often neglects or overlooks this relationship or gives overly-determinist explanations [40].

## 2. Contextual Information

This section of the paper will explore the context of this research, focusing on early school leaving in Spain and the characteristics of the province of Málaga. It is important to note that information at a regional level is limited due to data availability in Spain; however, an effort has been made to include existing information which can help readers gain a better understanding of socio-economic and educational conditions.

### 2.1. Early School Leaving in Spain and Policies Aimed at At-Risk Pupils

ELET has been a major issue for the Spanish Educational System historically [41], not only having an effect on education but also impacting social and economic spheres [42]. According to NESSE [43], those who leave early are more likely to experience unemployment, precarious jobs, or need social assistance. This is evident in Spain through the 19.9% unemployment rate among people with lower secondary education degrees compared to 8.8% of those with higher education degrees [44], as well as an estimated additional cost of €230,000 per student who leaves the educational system due to benefits such as health care and unemployment benefits [43–45].

In recent years there has been a decline in ESL numbers from 23.5% failing to obtain their diploma in 2015 down to 17.3 % by 2019 [44]. However, this still remains a high priority on both national and regional political agendas resulting in many policies being implemented towards reducing this number [46], which have so far failed to meet the 10 % target set by the Lisbon agenda.

Generally speaking, these policies focus on three main areas: (1) providing financial aid for families/individuals at risk, (2) setting up national plans starting from the primary level offering educational support and reinforcement, and (3) establishing specific programs within structure of education system that can provide environment better suited for attention diversity students at risk [47].

### 2.2. Málaga Region

Málaga (Spain) is a province with a diversified economy whose main industry is tourism, although other important industries such as agriculture, commerce and financial services also stand out. The total population of Málaga in 2022 is 579,076 inhabitants, with 278,086 men and 300,990 women; the urban core reaches 573,904 residents, while those dispersed are approximately 5172 people. The average age is 43 years old, with 19.9% under 20 years old and 18.6% over 65 years old; during the last decade, it has experienced a growth of 2.1%. Regarding foreign residents, there are 52,334 people, mainly from Morocco (18.3%). In 2021, there were 18,848 emigrations compared to 22,197 immigrations: additionally, 4396 births against 5487 deaths [48].

Data indicates that the labor market in Málaga is characterized by a significant percentage of people experiencing unemployment yet having one of the lowest unemployment rates in Spain. In 2022, 33,187 women and 22,777 men were registered as unemployed, resulting in a municipal unemployment rate of 20.9%. However, contracts registered were high: 141,079 for women and 153,737 for men; additionally, the number of indefinite contracts was 112,042 while 179,896 were temporary and 36,557 corresponded to foreign residents in the city. These figures suggest there are many opportunities available to find employment in Málaga, but at the same time, there exists a significant percentage of people seeking to employ their skills, so the competition may be tough [48].

In the field of education in Málaga, there are a total of 697 educational centers for the 2020–2021 school year. This includes 310 Early Childhood Education centers, 147 Primary Schools, 101 Compulsory Secondary Education (ESO) centers, 60 Upper Secondary Education centers, 42 Medium Grade VET centers and 49 Upper Grade VET centers, in addition to 39 centers dedicated to adult education. This situation demonstrates that there is a wide range of training available in the province, with different academic options for students from early ages up to 18 years old or more [48].

## 3. Materials and Methods

This study uses a grounded theory (GT) approach to explore the impact of ELET on mental health from the perspective of early leavers, guided by Charmaz's constructivist approach [49–53] and its informed version [54]. Qualitative methods were utilized in order to gain an understanding of participants' experiences with ELET and how this affected their mental health. In-depth interviews were conducted for this purpose. The data was then iteratively analyzed in order to identify patterns and construct categories of meaning related to the theme under examination. Robustness tests (exhaustive coding) were conducted throughout the process in order to ensure that the theory accurately reflected reality, but it has also been acknowledged a co-constructive role in achieving this outcome by the team in charge of the analysis [51,52].

It is important to note that the sample selection was part of a larger research project called YOUNG_ADULLLT (https://www.young-adulllt.eu/project/index.php, accessed on 10 April 2023), which was designed as a collaborative mixed-methods comparative study between Austria, Bulgaria, Croatia, Finland, Italy, Germany, Portugal, Spain, and the UK, focusing on lifelong learning policies for young adults—particularly those considered "vulnerable" such as NEETs and ELET—with the aim of critically analyzing current developments in LLL policies across Europe. Only results corresponding to our research question posed are presented here, but their belonging to said project determines how the sample has been collected. For this article, we have only used the qualitative part of data from the said project, analyzing information from interviews according to our research question.

### 3.1. Selection of Participants

In order to carry out participant selection, the snowball sampling method described by Noy [55] was used, through which participants accessed previously interviewed key actors. The specific criteria that guided the selection of our sample were as follows:

- Age between 18 and 35 years old (This criterion has been used following Whalter et al.'s [56] description of the "Young-Adult" approach in the project YOUNG_ADULLLT. The intention to increase the age range corresponds to a considered decision to explore longer trajectories. No minors were involved in the sampling because the project is focused on youth. Therefore, no permission has been obtained).
- Participants had a maximum qualification that corresponded to that stipulated under the concept of ELET and had a period in which they could fit into the ELET profile.
- Participants have returned to education and/or training policies.

### 3.2. Sample

The characteristics of the final sample, consisting of 10 males and 7 women, are presented in Table 1 below:

**Table 1.** Sample.

| | Code | Return Policy. | Nationality | Gender | Pre-Return Level | Socio-Economic Level | AGE |
|---|---|---|---|---|---|---|---|
| 1 | JA1_H_FP_EST | Vocational training (GM)—Administration | Spanish | M | Elementary Education | medium | 34 |
| 2 | JA2_H_FP_EST | Vocational Training (GM)—Computer Science | Spanish | M | Compulsory Secondary Education | medium | 27 |
| 3 | JA3_H_FP_EST | Vocational Training (GM)—Computer Science | Spanish | M | Elementary Education | low | 35 |
| 4 | JA4_H_FP_EST | Vocational Training (GM)—Computer Science | Spanish | M | Compulsory Secondary Education | low | 35 |
| 5 | JA5_M_FP_EST | Vocational training through open access (GM)—Pharmacy | Spanish | W | Compulsory Secondary Education | medium | 19 |
| 6 | JA6_M_FP_EST | Private and distance vocational training (GM)—Nutrition | Spanish | W | Compulsory Secondary Education | medium | 23 |
| 7 | JA7_M_EA_EST | Adult School | Spanish | W | Compulsory Secondary Education | low | 20 |

**Table 1.** *Cont.*

| | Code | Return Policy. | Nationality | Gender | Pre-Return Level | Socio-Economic Level | AGE |
|---|---|---|---|---|---|---|---|
| 8 | JA8_H_EA_EST | Adult School | Spanish | M | Elementary Education | low | 20 |
| 9 | JA9_H_EA_EST | Adult School | Spanish | M | Elementary Education | Medium-high | 24 |
| 10 | JA10_H_FP_EST | Vocational Training (GM) —Computing | Vltava | M | Compulsory Secondary Education | low | 20 |
| 11 | JA11_M_ESA_EST | ESA | Spanish | W | Elementary Education | low | 21 |
| 12 | JA12_H_FPE_MLG | Certificate of Professionalism | Spanish | M | Compulsory Secondary Education | low | 22 |
| 13 | JA13_H_FPE_MLG | Certificate of professionalism | Spanish | W | Elementary Education | low | 24 |
| 14 | JA14_H_FPE_MLG | Certificate of Professionalism | Spanish | M | Compulsory Secondary Education | low | 27 |
| 15 | JA15_M_ET_ALH | Workshop School | Spanish | W | Elementary Education | low | 27 |
| 16 | JA16_M_ET_ALH | Workshop School | Spanish | M | Compulsory Secondary Education | low | 24 |
| 17 | JA17_H_FPE_MLG | Certificate of professionalism | Peru | W | Compulsory Secondary Education | low | 30 |

### 3.3. Data Collection

Data were gathered through semi-structured retrospective interviews with open-ended questions. These queries were organized into three mains sections, beginning with a request for an overall description of the participant's academic journey, followed by intermediate inquiries about particular circumstances described in the initial section and concluding with some final questions that required the respondent to reflect on certain aspects observed and made explicit by the interviewer. This process was created following Charmaz and Belgrave's [57] guidelines, which explain each section's content while also emphasizing that, during this interview, it is necessary "to mentally categorize" information being given so as to include it in the conversation, highlighting the importance of having flexible conversations rather than rigid structures.

As Charmaz and Belgrave [57] argue, creating a comfortable atmosphere is essential for these interviews to be successful; therefore, interviews took place in familiar and discreet settings, allowing those interviewed to feel relaxed enough to share their stories. Audio recordings were taken of all interviews conducted to facilitate further analysis.

### 3.4. Data Analysis

Data analysis followed an inductive coding model described by Charmaz [50] using NVIVO to create and negotiate codes and categories: first by open coding and then undertaking an exhaustive coding until theoretical saturation was reached [58]. This software was used insofar as it allows for the creation and negotiation of codes and categories until theoretical saturation is reached [58], as has been identified by the literature [59] as a suitable tool for carrying out constructivist grounded theory analysis in a systematic way.

### 3.5. Final Codes and Categories Resulting from the Analysis

The codes and categories resulting from the analysis are:

1. ELET and its impact on self-esteem and mental health:
   a. Decrease in psychological well-being due to non-fulfillment of educational aspirations
   b. Decrease in psychological well-being due to loss of job aspirations
   c. Impact of continuous harassment on psychological well-being; (c.1) long-term consequences
   d. Academic stress
   e. Educational phobia associated with ELET

2. Social isolation and its relationship with ELET:

   a. Associated with disruptive behaviors
   b. Bullying
   c. Action for change
   d. Harmful behavior

3. Insufficient role of guidance services:

   a. Need for psychosocial support
   b. Academic orientation
   c. Post—ELET help
   d. Labor insertion
   e. Self—responsibilization for failure

### 3.6. Validity and Reliability

The importance of validity and reliability in grounded theory is contingent upon the quality of data collection and the thoroughness of the analysis process. In order to ensure reliable findings, it is imperative that high-quality data be collected. Additionally, validity and reliability are closely intertwined; when theoretical saturation [60] has been achieved through a validation process to review consistency between data concepts and theories, research reliability can then be ascertained. Theoretical saturation refers to the point at which no further information content can be added or obtained from a category—this study has successfully reached such an endpoint for all categories presented in its results.

### 3.7. Ethical Procedures

The ethical criteria applied in this research process adhere to those stipulated by the YOUNG ADULLLT project [61]. These standards are based on the Universal Declaration of Human Rights, the European Social Charter, and The Convention for Protection of Human Rights and Fundamental Freedoms. In particular, respect for human dignity, personal autonomy, welfare principle, and social justice, as well as confidentiality and information integrity, has been ensured throughout the study. Furthermore, professional responsibility has been maintained at all times [61]. In this way, ethics procedures were strictly adhered to: prior consent forms had been signed explicitly granting permission for recording purposes as well as informing participants who would have access and how their data would be used (including potential publications). All data has been anonymized accordingly—names have been changed along with places/locations blacked out—thus protecting those interviewed from any form of identification or recognition.

### 3.8. Data Availability

The YOUNG_ADULLLT research project seeks to make its findings available for use by other researchers as well as policy-makers who may be interested in using them for their own purposes. To this end, a web (https://www.young-adulllt.eu/publications/index.php accessed on 4 March 2023) [62] has been established upon completion of the study, which contains some of the information generated during the course of the research process along with detailed documentation about how they were created and used within each stage of analysis. This provides a valuable resource for future studies related to youth development issues within Europe while also promoting greater transparency around how such data are collected and utilized by researchers working on similar topics worldwide. The data can be accessed in two formats: working papers and policy briefs. Working papers providing an analysis of the data can be found at https://www.young-adulllt.eu/publications/working-paper/index.php (accessed on 4 March 2023), while policy briefs outlining the implications of the study are available at https://www.young-adulllt.eu/publications/policy-briefs/index.php (accessed on 4 March 2023).

## 4. Results

The results of the research are presented below, having been organized around the core categories resulting from the analysis and their associated codes. This is intended to provide greater transparency about the process of analysis and its form.

### 4.1. ELET and Its Impact on Self-Esteem and Mental Health

The initial core category identified through the analysis pertained to various circumstances that had an effect on the self-esteem and mental health of young leavers. These moments and conditions were unequal, thus necessitating each to be presented individually:

(a)   Diminished psychological well-being[4] due to non-fulfillment of educational aspirations.

One of the categories identified through analysis was a decrease in psychological well-being among interviewees due to their failure to realize educational aspirations. Generally, life course accounts of young people interviewed revealed an association between ELET and deteriorating mental health; however, this phenomenon is only observed amongst those who left after the 2008 financial crisis and is more pronounced among those who departed during Upper Secondary Education with higher expectations for education.

> *"When I was a child, studying was basically a duty. It was expected of me to get good grades, and the pressure of not achieving this could be quite overwhelming. It felt like I was never going to reach the set goals and that it would never be enough. I felt the pressure".* JA10_H_FP_EST

> *"I had a difficult time after leaving school. I think it even led to depression. Since I was a child, I had to be the "pioneer" of my family since no one had ever obtained a degree. My childhood ambition was to accomplish something great, no matter what it was".* J05_M_FP_EST

In these fragments, we can observe how ELET is linked by respondents to a decrease in self-esteem and a depressive process due to the difficulty for these individuals to meet their educational aspirations or those that were acquired within their family environment. In some ways, they suggest that ELET results in the loss of part of one's identity. When young people leave school, they report losing an aspect of themselves as students, which had been constant since they began attending Spanish schools at around three years old; this often leads to an imbalance exacerbated by familial pressure. These findings are aligned with other studies, which point out that conditions associated with transitioning into adulthood—such as uncertainty—may have psychological effects and even affect the rate at which someone constructs their identity [64]. This brings us indirectly face-to-face with the impact yo-yo trajectories between ELET and return could have on its protagonists.

Furthermore, in addition to the initial unease caused by ELET in the psychological realm, there is a lingering shadow of the past that some interviewees claim has an enduring effect on them. This shadow appears as an intrusive and persistent thought which erodes their self-esteem and leads them to ponder what might have been if they had invested more resources into education or considered alternative scenarios.

> *"My mind was in a whirl, and I could not help but think: -If only I had done more, I could have achieved more. But, that's life, right? I like to think that I could have still made it, even if I was not able to finish all my courses in the summer. I'm not sure, but I just could not shake these thoughts".* J05_M_FP_EST.

(b)   Decreased psychological well-being due to a loss of job aspirations.

Closely related to the preceding category, an emergent one from the analysis alludes to a decrease in psychological well-being due to a diminution of job aspirations associated with an absence of educational qualifications. As with the previous category, this one displays different effects depending on respondents' age; it appears much earlier among those who left after the 2008 financial crisis.

Hence, a common theme in the interviews of young people is the direct relation between losing self-esteem as a result of ELET and being associated with the concept of

NEET. This can be clearly seen from comments made by interviewees JA14_H_FPE_MLG, JA17_H_FPE_MLG, JA5_M_EST and JM16 M ALH, who allude to this notion and how it weighs heavily on their mental well-being.

*"The feeling of being a NEET can be incredibly overwhelming. It's a sense of helplessness that is difficult to combat, no matter how hard one might try. However, I believe that nothing in life is permanent. Everything changes over time, nothing is eternal. Everything is fleeting". JA17_H_FPE_MLG*

Despite this, as we can observe in the preceding excerpt, a considerable number of interviewees express reluctance to identify with this concept, attempting to reevaluate whether belonging to the group is an identity association or a one-time occurrence. Although within the general discourse of young people, this situation is associated with individualized failure and thus adds another layer to the "wound" of ELET.

*"I can relate to the feelings of being a NEET, as I too have experienced them. However, it can be difficult to break through the obstacles that we put in our own way, and even when we do, people can put them back in our paths". JM16_M_ALH*

Furthermore, among older respondents' discourse, there is a relation between the decision to leave early and long-term unemployment. Most of them emphasize that this association—ELET leading to unemployment which then leads to hopelessness in the labor market resulting in depression—did not exist prior to the crisis; it was only after its onset that it began manifesting itself. This suggests a much later individualization of failure, although both groups experienced it at around the same time.

*"When the crisis in the construction industry arrived, it had an especially strong impact here. There was a high demand for work, yet I was unable to be hired due to my lack of a degree and experience as a professional. I was left feeling helpless and hopeless. This feeling of helplessness caused a psychological downturn, and I ended up claiming unemployment. After two years of being unemployed, I struggled to get back on my feet. This further contributed to my depression, and I began to feel like I would never be able to achieve anything with my life". JA3_H_FP_EST*

(c)　The impact of continued bullying on psychological well-being: a long-term wound

Another category that emerged from the analysis is bullying and its effect on psychological well-being. It is noteworthy to observe a group (JA06_M_FP_EST, JA09_H_EA_EST, JA11_M_ESA-EST, and JA16 M ALH) who associate continuous bullying with psychological distress leading to "withdrawal" from school which eventually culminates in leaving.

*"At school, I often found myself not wanting to go to class, so I began to skip them. Being a very quiet person, I was often subjected to bullying from my peers. This took a toll on me mentally, affecting my studies as I was always fixated on what was said to me, regardless of whether it was good or bad. As a result, I found it increasingly difficult to concentrate on anything else, and this further hindered my studies". JA11_M_ESA_EST*

It is of particular importance that many of the young people interviewed take responsibility for their involvement in bullying. This is exemplified by JA16_M_ALH, who stated that "if she had been more assertive with regards to tackling bullying", she might have completed her studies.

*"I think I should have taken more initiative to stand up to the bullying and complete my Upper secondary education. I regret not having done so, and I feel that if I had, I might have gone on to pursue a degree". JA16_M_ALH*

Bullying is currently one of the most psychologically damaging issues in education, and unfortunately, its victims often suffer from long-term psychological effects and a decline in mental health [65]. The interviewees in this study reflect this relationship, with many of them calling for increased attention from schools that they believe are not doing enough to combat these consequences. However, their claim is based more on ending

bullying rather than on the belief that educational guidance and psychological support can improve overall well-being and health.

Furthermore, although many of them necessitated psychological treatment or confessed to needing it, most exhibited a certain reticence toward attending therapy. This appears to be rooted in the erroneous assumption that "one must resolve their own issues". This further reinforces the individualization of experiences related to ELET.

> *"After spending around a year in a state of uncertainty, I decided to see a psychologist in order to gain insight into the issue of bullying at school which I was facing. The guidance I received enabled me to enroll in a workshop school and this gave me the opportunity to reflect on my situation. It was a difficult period, as I felt lost and confused. However, after two sessions with the psychologist I was able to gain clarity and had a much better understanding of my situation. It was more me and my mind that changed than anything else. Now, I'm in a much better place and have been able to move on from my previous difficulties". JA16_M_ALH*

In relation to this, it is interesting that some of the interviewees associate themselves with patterns determined by a traditional view of gender, pointing out that this self-responsibilization can lead to worse social adaptation in high schools and result in situations of bullying which may culminate in experiences of ELET. The ways that respondents explain "being" or "inhabiting" the school space as a man or woman refer us to an outdated notion of masculinity where one must avoid displaying any kind of signs of weakness, and hyper-femininity, which refers to "groups of popular girls" who identify themselves as "fat" or "shy".

> *"When I was in third year of ESO, my classmates started comparing me because I was chubby, and they were a size 38. I felt like I had to be like them and asked myself why I couldn't be. I was a size 46 in trousers. I tried to focus on losing weight and going to the gym as much as I could, but this was unfeasible during high school. On top of that, I wanted to start dating, so it was a very difficult time. At school, nobody paid attention to or liked a chubby girl. Fortunately, I have been able to lose weight since then". JA06_M_FP_EST*

> *"As a student, I didn't receive much attention at school. People would say:—Look at this little fool, I don't know him.—This was likely because I was so involved in religious matters from a young age and was very Catholic. People would say:—Look, the religious guy who's good for nothing.—Nowadays, I think it is mainly boys who provoke bullying. From their perspective, they might think:—I know this guy, let's bully him because he won't get angry.—I never wanted to cause that much trouble".JA09_H_EST*

(d)    Academic stress and educational phobia associated with ELET.

The final category that emerges from this core category is related to the relationship between ELET and academic stress and phobia in regard to the educational environment. This was reported by some of the students who prematurely left school. They explain that a major factor behind their decision to leave was an intense level of academic pressure stemming from expectations imposed by teachers for each subject, which were often too difficult or unclearly defined for them to meet, leaving them feeling as if they had no control over their situation. This phenomenon has been identified among all interviewees who have departed from Upper Secondary Education.

> *"It was incredibly frustrating; I couldn't keep up with the amount of work that was expected of us. I felt like I was completely out of my depth and had no idea what was going on. It felt like the situation was changing and I didn't know how to take control of it. I was completely blocked; I felt like I had reached a dead end and could not comprehend the material".JA05_M_FP_EST*

Furthermore, as observed in the preceding section, some of the students surveyed reported developing a sense of fear or apprehension when confronted with a school environment that they felt unfamiliar with and ill-equipped to handle.

*4.2. Social Isolation and Its Relation to ELET*

The second core category pertains to a tendency towards social isolation that is linked to ELET. This has been included for two reasons: firstly, due to its saturation in the analysis and thus, given its prevalence in the discourse of young people, it should be acknowledged; secondly, because there is an established connection between ELET, isolation, and a detriment to mental health—despite seeming unrelated at first glance when planning the objective of our analysis.

Social isolation has been identified in the literature as a pervasive issue in modern society. Recent major structural, social and cultural changes have significantly altered how people form relationships, leading to an increase in social isolation. Individuals often prioritize their own needs over connecting with others, resulting in fragmentation of experience and the development of individualism, which further weakens supportive community ties. This leaves individuals feeling alone and searching for self-gratification [66].

Culturally, there has been a tendency to associate youth with sociability. However, there is an increasing interest in exploring how loneliness is related to adolescence and youth; this period of life involves a dual process of detachment with parental figures, as they are distancing themselves while simultaneously seeking new relationships among peers. According to Pretty et al. [67], loneliness in young people may be caused by the inability to satisfy their needs for peer or intimate relationships, leading them towards feelings of isolation from society and lack of community support. Additionally, some youths may experience discrepancies between their expectations regarding social skills and opportunities that can result in periods of seclusion if not adequately addressed [66]. Recently these phenomena have been referred to as "freeter", "otaku", or "hikikomori" within Japanese societies [68].

Our interviewees JA06_M_FP_EST, JA15_M_ET_ALH, JA11_M_ESA_EST, JA12 H FPE MLG, and JA13 H FPE MLG allude to a situation of social isolation that is directly linked to their decision to discontinue their education prematurely.

*"When I left school, I was all alone. I had to leave my friends and the school community I was part of, so it was a complete break with everything I had been used to".* JA06_M_FP_EST

However, the ways in which loneliness and isolation are referred to, as well as the reasons behind them, are diverse. Each of these is briefly explored below:

→ Social isolation due to disruptive behaviors: Individuals who refer to social isolation due to disruptive behaviors in the school environment often explain their loneliness because of a series of disruptive actions that led to them leaving and, ultimately, isolating them from their peer group:

*"I was often viewed as the "cool one" because of my bad behavior, but I always had good relationships with others. Unfortunately, I could come across as annoying, which caused people to shut me out. This left me feeling isolated and lonely, which was very difficult to cope with".* JA15_M_ET_ALH

→ Social isolation due to bullying: As evidenced in the preceding section, numerous interviewees have alluded to social isolation due to bullying, which has ultimately resulted in them withdrawing from the educational environment:

*"Due to feeling uneasy in class, I was afraid of a certain girl. This made the other students start to have a negative opinion of me, making me feel isolated and unwelcome. I was left alone with no one to turn to".* JA11_M_ESA_EST

→ Social isolation as an action for change: In these cases, interviewees reported a rupture with their peer group following ELET, which was an intentional decision to better their circumstances and habits in response to a landscape of social behaviors that had been contributing to an identified unhealthy and unhelpful situation:

*"I've been living in this neighborhood with my friends all my life, and since I started the new module of VET I've noticed a lot of young people in need of help, we need it, if*

*anybody can help us. My friends and I used to be in the same group, but I decided to distance myself and move out. I saw them going down the wrong path and I didn't want to be part of it. I wanted to be in a more peaceful environment, and that's why I now stay at home with my girlfriend in a quiet house. I come home, have dinner, and then go to bed". JA12_H_FPE_MLG*

→   Social isolation due to unhealthy behaviors: Some interviewees alluded to social isolation following ELET, which was not indicative of depressive patterns or absolute seclusion; rather, it was highlighted as a period characterized by low activity levels, prolonged periods spent at home and the use of substances detrimental to health.

*"I was confined indoors all-day, chain-smoking cigarettes or weed... I spent a lot of time at home too. I would wake up around three in the afternoon, smoking three joints, and remain in my house until evening. The tedium of day after day was hard to bear, and I found it difficult to quit". A13_H_FPE_MLG*

→   Social isolation because of a loss of self-esteem related to ELET: This referred isolation appears to be the most severe of those described, as it is associated with depressive processes and a situation that resembles "social phobia" due to the shame experienced by individuals affected by ELET.

*"I was alone and felt isolated. I would get dressed up, saying I was going out, yet I couldn't bring myself to do it. I would take off my make-up and start crying, feeling judged by those still attending school, by my own friends. Every morning I woke up in my pajamas, feeling like I was doing nothing. I felt like a housewife, stuck in a routine of getting up, walking the dog, doing the washing and cleaning the house, followed by my mother coming for lunch. This was the worst thing for me, and I envied those at school, wishing I was on their wavelength". JA05_M_FP_EST*

*4.3. The Insufficient Role of Guidance Services*

The third core category indicates that the respondents felt a need for more personalized guidance services with comprehensive attention to the ELET process. In Spain, all schools have an Orientation Department or Unit, which is an internal service of the school and consists of specialized personnel such as pedagogues, psychologists or psycho-pedagogues, social educators and teachers trained in therapeutic pedagogy and hearing/language specialists [69]. However, almost unanimously, interviewees (JA04_H_FP_EST, JA05_M_FP_EST, JA06_M_FP_EST, JA09_H_EA_EST, JA10_H_FP_EST, JA11_M_ESA_EST, and JA16_M_ALH) alluded to the fact that these services were insufficient and inefficient.

Against this backdrop, the interviewees highlighted the need for increased involvement from school guidance services. They described these services as being largely absent and ineffective, with minimal interaction between pupils and staff members and almost no intervention in cases of ELET.

*"In my high school, I felt blocked, and I wasn't sure why. I wish I had gone to a psychologist for help in understanding it, but the school was so large that the counselors were too indifferent to me to be of any assistance". JA05_M_FP_EST*

*"The guidance counsellors ... let's say that they didn't care about unruly children". JA10_H_FP_EST.*

This claim focuses on three main aspects: more psychological support, better educational guidance and post-ELET job counseling.

*"If they had provided us with a bit of orientation to the labor market, I may have been more informed. Maybe, I wouldn't leave. However, I cannot say for certain as I may have not taken the advice to heart". JA04_H_FP_EST*

*"When I asked for guidance about my studies, I was left feeling even more confused. Instead of helping me find what I wanted to do and where I should go, the person gave*

*me conflicting advice. This only added to the confusion I already had in my head, and I didn't find any help with the orientation process". JA09_H_EA_EST*

However, as we discussed in the section on bullying experiences, many students take responsibility for their failures and are thus reluctant to seek help from these services, particularly when requests are voluntary, and no follow-up is conducted.

*"There was a counsellor, but of course, maybe I was also reluctant to tell my things to a stranger. And, with the fact that it was voluntary counselling, I had to attend when I thought I should, so I didn't go". JA04_H_FP_EST.*

## 5. Discussion

In this research, we explored the perception of early leavers about the relationship between mental health and ELET. The results of our study help to further our understanding of this relationship and enable us to formulate mid-range theories using grounded theory.

In contrast to research that demonstrates a direct relation between ELET and a decrease in mental health, our data have revealed that while this relationship does exist, it is not necessarily the immediate consequence of leaving school. Consequently, the connection between school departure and psychological well-being among young people should be comprehended from its bidirectionality; where the detriment that our respondents highlight to their mental health as a result of ELET should be considered both an antecedent and outcome of this process.

Furthermore, according to our interviewees, we must consider the role that bullying plays in the decision-making process of young people when leaving education, as fear, isolation, and general distrust caused by bullying can lead to a gradual withdrawal from educational environments. Despite this, there has been limited research on the relationship between ESL and bullying both nationally [70,71] and internationally [72–80]. Although these studies do not specifically address this relation directly, they suggest that there is indeed a direct link between them. Therefore, further empirical research needs to be conducted in order to gain insight into how these dynamics interact with one another, which could potentially explain why some students experience truancy or early school leaving due to factors such as bullying. Our study provides an opportunity for those affected by such conflicts to voice their experiences which may help us better understand what lies at the root of these issues.

As stated in the theoretical framework, young people at this stage are exposed to various stressors of a social, financial, and academic nature that can have a negative impact on their mental health [81]. According to the literature, lack of time for meeting academic obligations, excessive workloads and material to study as well as an overload of academics were found to be the most common sources of stress. Our results aligned these findings by showing that students who leave school prematurely may also be subject to such pressures, thus, indicating that academic stress is not only experienced by those with higher grades or more competitively minded individuals. Furthermore, our results help us gain further insight into this process by illuminating stages that may generate high levels of pressure—in the case of our sample occurring during Upper Secondary Education—where young people interviewed associate their distress with difficulty in following an educational routine whose implicit rules they do not fully comprehend. It is important to note, too, that our outcomes suggest prolonged exposure to such academic strain culminating in ELET could potentially lead them towards developing some kind of "phobia" concerning school environments later on in life.

On the other hand, our research has revealed that ELET can have a negative impact on mental health among young leavers. Our sample identified three main circumstances: depression, social isolation, and decreased self-esteem. These effects had already been noted in the previous literature [9–11]. However, this study has allowed us to further explore the mechanisms behind this relationship by identifying loss of labor and academic aspirations as highly influential elements. Furthermore, we have determined conditions dependent on current socio-historical situations which contribute to a decrease in mental health amongst

young leavers. In particular, it is noteworthy how much pressure NEET labels exert upon their self-concept as well as personal and work relationships. This necessitates further investigation into the potential for unintended consequences of economic strategies and indicators. Informed by a neoliberal perspective, such measures may have an effect on young people who depart from traditional life courses.

Moreover, our analysis revealed an unexpected relationship between ELETs and social isolation. Social isolation and loneliness have not typically been associated with youth [66]. However, terms such as "freeter", "otaku", or "hikikomori", which refer to phenomena related to loneliness among young people, are increasingly being explored. These studies are largely influenced by the societies in which they were developed; mostly located in Asian countries, which have socio-economic and cultural characteristics that mean that youths' experiences differ greatly from those of Western youth [68]. Although these concepts have yet to be linked with ELETs', the literature suggests a close association with the NEET phenomenon. Kaplan and Gangestad [82] suggest there may be a connection between hikikomori and NEETs, noting their tendency to prefer immediate rewards over long-term investments [68]. This view of both phenomena can lead to hyper responsibilization of the loneliness experienced by individuals who often blame themselves for processes closely related to social exclusion—structural problems that our study shows they did not cause. Our results point towards an alternative perspective where many cases of social "withdrawal" may represent strategies designed to improve individual situations through personal tactics used negotiate structural tensions imposed on them, supporting Furlong's [83] thesis closer to our perspective. By this, our study provides an opportunity for those affected by such conflicts to voice their experiences which may help us better understand what lies at the root of these issues.

Finally, it is noteworthy that our study reinforces the idea previously highlighted in theoretical frameworks that a large number of young people have great difficulty assessing their mental health [21,26,84]. Our findings indicate that often those who leave school prematurely despite recognizing certain damage to their mental health process take responsibility for it by explaining from an individual level perspective how they must take charge and solve their problems. In doing so, they demonstrate a certain resistance to using psychological care. This is even more significant when considering the distrust that these young people show towards school counseling services, leaving us with a scenario where multi-dimensional interventions are necessary, not only working on psychological or social factors but also fostering greater confidence in these services.

## 6. Implications for Policy Making

Based on the findings of this research, policy-makers should consider a range of possible actions to address ELET in order to improve mental health among young people. First, policies that focus on addressing bullying in school settings are needed, as it has been identified as an important factor influencing ELET decisions. These could include initiatives such as anti-bullying campaigns or programs aimed at creating safe learning environments for all students.

Second, measures should be taken to reduce academic stressors by providing more support and guidance for students who are struggling with their studies, including those facing financial difficulties or other challenges related to a socio-economic status that can impact their educational outcomes. This could involve increasing access to tutoring services or offering additional resources such as study materials and mentorship opportunities for those who need them most.

Thirdly, in line with the analysis of the responses of our interviewees, greater attention needs to be paid to the potential unintended consequences of economic strategies informed by neoliberal perspectives, which may have negative impacts on young people's lives if not addressed appropriately. Policies must therefore consider how these dynamics interact with one another when formulating plans designed specifically for youth affected by poverty and social exclusion due to early leaving education processes. Finally, it seems to be a need

for multi-dimensional interventions that provide psychological care while fostering trust in counseling services so that young people feel comfortable seeking help when necessary without feeling ashamed or judged.

## 7. Conclusions

This study aimed to develop intermediate-range theories that would help gain a better understanding of the relationship between ELET and mental health. Although the sample size was sufficient according to qualitative parameters, it was limited and could not be generalized; thus, the results should be taken as a hypothesis rather than disregarded as they represent an important advancement in knowledge regarding youth mental health. Through this study, we explored the ELET-mental health relationship and contributed to developing new lines of research concerning phenomena such as bullying, loneliness, and social isolation during youth. From a social psychology perspective, we observed how certain neoliberal values are negatively impacting young people's mental health by internalizing self-responsibilization for failure and subjecting them to labels such as NEET, which further reinforces this idea while undermining their self-esteem. In conclusion, our proposed objectives were fulfilled; it is essential to continue exploring these newly presented mid-range theories and find ways to carry out interventions that can minimize the associated problems discussed herein.

## 8. Limitations of the Study

The present study has certain limitations that should be taken into account when interpreting and generalizing its results. Firstly, due to the use of a grounded theory, it is difficult to make any claims regarding generalizability. The sample size was also limited in terms of geographical area (Málaga province) as well as age range (18–35). Furthermore, this research focused on Spanish participants only, which may limit the applicability of these findings across different cultural contexts. Finally, given that this study used retrospective data collected through interviews with former ELET students, there is potential for recall bias which could have impacted their responses and interpretations. Memory gaps or errors, re-interpretation of experience over time, and other issues related to the use of retrospective data should be taken into consideration when interpreting the results of the present study.

## 9. Further Lines of Research

Given the limitations discussed above, further research should focus on expanding sample sizes by including more countries and cultures while incorporating quantitative methods such as surveys or experiments in order to explore causal relationships between ELET policies and mental health outcomes among young adults. Additionally, longitudinal studies would provide valuable insight into how experiences during early life transitions can shape mental health trajectories over time, thus helping inform public policy decisions related to youth education and training programs at both national and international levels.

**Author Contributions:** Conceptualization, L.M.G.-P. and M.A.G.; methodology, L.M.G.-P.; software, L.M.G.-P.; validation, L.M.G.-P.; formal analysis, L.M.G.-P.; investigation, L.M.G.-P.; resources, L.M.G.-P. and M.A.G.; data curation, L.M.G.-P.; writing—original draft preparation, L.M.G.-P. and M.A.G.; writing—review and editing, L.M.G.-P.; visual-ization, L.M.G.-P.; supervision, L.M.G.-P.; project administration, L.M.G.-P.; funding acquisition, L.M.G.-P. All authors have read and agreed to the published version of the manuscript.

**Funding:** This research is associated with the project funded by H2020, YOUNG_ADULLT. But funding has being fully assumed by the authors. The APC was funded by MDPI through reviewer vouchers, University of Granada IOAP program, and partially paid by authors.

**Institutional Review Board Statement:** Not applicable.

**Informed Consent Statement:** Informed consent was obtained from all subjects involved in the study.

**Data Availability Statement:** A web (https://www.young-adulllt.eu/publications/index.php, accessed on 4 March 2023) [62] has been established upon completion of the study, which contains some of the information generated during the course of the research process along with detailed documentation about how they were created and used within each stage of analysis. This provides a valuable resource for future studies related to youth development issues within Europe while also promoting greater transparency around how such data are collected and utilized by researchers working on similar topics worldwide. The data can be accessed in two formats: working papers and policy briefs. Working papers providing an analysis of the data can be found at https://www.young-adulllt.eu/publications/working-paper/index.php, accessed on 4 March 2023, while policy briefs outlining the implications of the study are available at https://www.young-adulllt.eu/publications/policy-briefs/index.php, accessed on 4 March 2023.

**Conflicts of Interest:** The authors declare no conflict of interest.

## Notes

1   According to the World Health Organization [1], mental health is "a state of well-being in which every individual realizes his or her own potential, can cope with the normal stresses of life, can work productively and fruitfully, and is able to make a contribution to her or his community".

2   The term 'early leavers from education and training (ELET)' is used to refer to individuals aged 18–24 who have completed, at most, lower secondary education and are no longer engaged in any further educational or vocational pursuits. This statistic is determined using data collected by the Labor Force Survey (LFS). The figure for this indicator is calculated by dividing the number of people within that age group fitting these criteria with those surveyed as part of the LFS (European Comission, s.f) [4].

3   According to Ryff and Keyes [18] and Ryff and Singer [19], psychological well-being is a factor of optimal functioning that reflects an individual's capacity for self-acceptance, positive relationships with others, autonomy, environmental mastery, purpose in life, and personal growth. It involves the ability to express feelings of empathy and affection for other people as well as the realization of one's potentialities through continuing development over time.

4   Psychological well-being is a broad construct that encompasses subjective experiences of life satisfaction, positive affect and self-realization [63].

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
