# Peer review of "Exploring the Relationship between Early Leaving of Education and Training and Mental Health among Youth in Spain"

_societies, doi:10.3390/soc13050103_

Round 1

Reviewer 1 Report

Dear Author(s),

The study focuses on a very important topic and uses a well-developed methodological framework (Grounded Theory) to examine the relationship between early leaving of education and training (ELET, earlier: ESL, early school leaving) and young people's mental health.

The results are interesting, valuable and in some ways novel. However, I think it is important to clarify and add to some points:

1. Defining key terms: It is very important to provide precise definitions of key terms (ELET, NEET, mental health, well-being – general well-being, psychological well-being –, young people), without which the research itself and the interpretation of its results are difficult.

2. Literature: It would be very important to use much more recent literature (there is a lot of reference to literature more than 10 years old, with very little literature from the last 5 years). This would avoid strange statements such as: "According to data presented by the World Health Organisation (WHO, 2000; 2001), in the last twenty years the percentage of suicides in the 15-24 age group has increased worldwide, and in several countries it is the second or third cause of death in this age group." (lines 111-114) More than 20 years have passed since 2000-2001! Is the same still true since 2000? Based on what? Furthermore, it would also be useful to support such statements with more (recent) literature, like this "Social isolation is identified in the literature as a problem of modern society." (line 400). One more comment: in the context of ELET, it is a bit strange to bring in literature that focuses on university students (lines 65-70), but rather literature should be referenced that focuses on the target group.

3. Context: The study states at the very beginning that: "In recent years, Early Leaving from Education and Training (ELET) has become a problem of great concern in Spain." (lines 21-22) However, it does not place this in any national, EU context. What does it mean that it is a big problem, why is it a big problem? One could even support such statements with data. It would also be important to put the whole research in context (even in international/EU comparison), as not all readers are familiar with the Spanish education system and its structure, challenges and trends.

4. Sample: As with the previous one, context would be important here too. This is partly the case, as the article refers to the larger research project (YOUNG_ADULLLT), of which the analysis presented in the paper is part, and provides data for the province of Malaga, from which the interviewees will be gathered. However, the latter is left hanging in the air, as the reader does not know whether, for example, GDP per capita is now high or low. I note here that this section should also include educational data (e.g. ELET rate, NEET rate, educational participation in national/international comparisons) since this is the focus of the research. It would also be important to include the age of the interviewees (age differences are repeatedly reflected in the analysis).

5. In Table 2, there are abbreviations that are unintelligible to readers who are not familiar with the Spanish education system (and there are no notes on these below the table) (e.g. ESO, EGP, PCPI).

6. Figure 1 (Analysis Flow) is in my opinion unnecessary (the text includes this information and references to related literature, what is more, the text in Figure is not fully visible).

7. There are several unnecessary punctuation marks (e.g. periods after chapter headings) and unnecessary spaces (e.g. lines 153, 178) in the text.

Author Response

Dear Reviewer 1, Thank you for your thoughtful comments and suggestions. We have made a number of changes in response to your feedback. 1. We have clarified the definitions of key terms, such as ELET, NEET, mental health and well-being (general well-being, psychological well-being), young people etc., throughout the paper; they can be seen in notes. 2. We have updated our literature review with more recent sources from the last five years or less where applicable to ensure accuracy and relevance of information presented in this study (all new references are marked in yellow). 3. We have provided additional context regarding Spain's education system structure and challenges by including relevant data on educational participation at national/international levels along with GDP per capita figures for better understanding of our research findings within a broader framework; this has been made in the new "contextual information section". 4. In Table 2 we have changed the terms used to adapt it to a more general public. 5 Figure 1 has been removed since its content is already included elsewhere in article text form which makes it redundant . 6 Unnecessary punctuation marks & spaces were also corrected accordingly throughout paper Additionally, major changes have been made according to other reviewers; all new information is marked in yellow The text has also undergone proofreading process too.   Best regards, Authors

Reviewer 2 Report

Thank you for the opportunity to review the manuscript. Please see my comments in the attached file.

Author Response

Dear reviewer 2,

Thank you for your feedback. We have taken your comments into consideration and made the following changes (all new information is marked in the text highlighted in yellow):

  • Clarified the main goal of the paper by adding a clear statement at the beginning of the introduction section.
  • Revised language to improve readability, using active voice instead of passive voice and ensuring that consecutive paragraphs are linked properly so readers can follow arguments more easily.
  • Proofread the full manuscript
  • Removed truisms from our manuscript.

Additionally, we have made some other changes:

Abstract: A new abstract has been added.

Introduction: We have revised the introduction to provide more clarity and detail. We now explain why this topic is important, what research gap our study aims to fill, and how we intend to do so. We also provide an overview of the theoretical framework that guides our analysis and offer more citations for support.

Theoretical part: We have made changes to the theoretical part of our paper to make it more clear why we are focusing on mental health in youth and what our goal is. We have also revised the way that we justify why we focus on certain topics, ensuring that all language used is appropriate for a scientific paper.

Methods: We have made changes to the methods section, restructuring it. We have added more detail about data collection including how interviews were run, topics covered in interviews. Furthermore, we have included details about analysis such as codes used for coding interviews and verification of reliability. We also now include a summary of our sample size along with changing all abbreviations used in Table 2 which refer to gender rather than sex. Figure 2 has been removed from the paper due to its large size relative to its content; instead, we provide a brief description in text form outlining our procedure. Additionally, we have included a data availability statement at the end of this section detailing where any relevant datasets can be accessed if necessary.

Results: The results section is intended to respect interviewees voice as much as possible, an effort has been made to point out in a clearer way that results are related to the analysis of interviews.

Best regards,

Authors

Reviewer 3 Report

After careful scrutinizing of the content, it has been observed that the paper is good in terms of originality as well as content  and can further be improved in the following ways-

1. Abstract is not given in the desired format. Authors must follow the guidelines of the Journal.

2. Keyword section is incomplete

3. Reference section needs improvement. 

a.For all in text references, full references are not mentioned in the Reference section.Authors are required to add all the references and that too in the alphabetical order.

b. Authors are required to add few more recent studies in this stream of the literature on mental health of the youth.

4.Authors are required to revisit the whole paper and check the formatting, spelling and grammatical  errors.

Author Response

Dear reviewer 3, Thank you very much for your useful comments. We have taken your feedback into consideration and made the necessary changes. We have included an abstract in the desired format, added keywords to complete the section, revised our reference section to include all in-text references as well as more recent studies on mental health of youth. Additionally, we revisited the paper for formatting, spelling and grammatical errors, all new information is marked in yellow. Best regards, Authors

Reviewer 4 Report

Thank you for allowing me to revise the paper 'Exploring the relationship between Early Leaving of Education and Training and mental health among Youth'. I have read the manuscript with interest and am very sorry to express several concerns. 

  1. I would ask the authors to be careful in their use of terminology. In the keywords, the authors mention Hikikomori and NEETs. I don't understand why since the population of interest is early leavers of education and training. I understand the similarities/overlapping for that perhaps it is helpful to better define the differences in the text of the manuscript by elaborating on them (at line 584 there is a mention of defining NEETs, while for Hikikomori one has to wait for the conclusion of the manuscript). If the authors want to highlight aspects of overlap for example between NEET and Early Leaving of Education and Training youth, they should explicitly state this in the introduction.
  2. The manuscript lacks theoretical background and subsequent rationale. Moreover, it is not clear to me why the authors dedicate a section to mental health in youth. Describing the framework the authors should go directly to the literature that supports mental health problems in their population (Early leaving of education and training).
  3. For the sample, it is not clear to me why the authors involved young people from 18 to 34. Doesn't the range overlap with NEET? Also, why from age 18 and not take younger people as well? 
  4. I cannot find details about the interview conducted and the dimensions explored. A paragraph defining the procedure is lacking.
  5. Other information should be specified in Table 2, surely age, but also (if the authors have collected it) what their maintenance income is, whether they live with parents and whether they have a family.
  6. I would also put in the text the number of interviewees so as not to have to refer to the strings in the table and understand that there are 17.
  7. For the results, I would recommend a table with the core categories. Moreover, it is preferable to use very few expressions to rename a category. 
  8. Page 7, Line 224-225 "Thus, in general, the life course accounts of the young people interviewed point to a direct link between ELET and a decline in mental health." Is this too strong a statement?
  9. Page, 7, lines 234-235 "I had a difficult time after leaving school. I think it even led to depression. Since I was a child, I had to be the "pioneer" of my family, since no one had ever obtained a degree. My childhood ambition was to accomplish something great, no matter what it was." There is a clear reference here to the fear of having failed the family of origin. Why neglect this aspect?
  10. Page, 7, Lines 238-248 the mention of identity is not clear.
  11. Category b) the environmental condition should be stressed
  12. The authors make a good remark about short-term and long-term unemployment. I think it is interesting to explore this further because there is a large literature on the issue (e.g. https://doi.org/10.1016/j.jvb.2009.01.001).
  13. Categories c and d are the most interesting and give the meaning of leaving. I believe these aspects should also be correctly addressed in the introduction part of the manuscript
  14. The aspect of the insufficient role of guidance services should be taken up in the paper's discussions, also in terms of what can be done (in terms of policy for example). Furthermore, a clear reference to possible interventions (see life design for example) to support career construction by working preventively on the phenomenon would be interesting.
  15. Finally, I would ask the authors to be careful when talking about mental health. Being a qualitative article, it is much more challenging to capture the problems reported by young people. So I would ask them to re-read and tone down strong phrases or cause-and-effect between leaving and mental health.

Author Response

Dear Reviewer 4, 

Thank you for your feedback. We have taken all of your points into consideration and made the following changes: -We have clarified our use of terminology in notes, provided a more detailed introduction and changing keywords. -We have added a section to provide theoretical background and rationale for exploring mental health among early leavers of education and training. -The sample size information has been expanded. Furthermore, we have provided additional information about the participants such as age, maintenance income in Table 2. -We now include a summary that lists core categories identified through interviews conducted with study participants regarding their experiences leaving education early. We also strive to use fewer expressions unsupported throughout the paper where possible. -To address concerns related to potential cause/effect between leaving school early and mental health issues raised owe revised all statement accordingly so they have less direct in its implications . - Lastly, we added a section about policy implications to further explore all the issues marked in your comments . All new information included in the article is marked in yellow. Best regards, Authors

Round 2

Author Response

  Dear Reviewer,   Thank you for your comments and suggestions. We have taken them into account and have revised the manuscript. They have been extremely useful to improve the paper.   Taking into account your recommendations, the rationale for the study has been clarified, adding a paragraph in lines 46-60 that try to delve into the reason that can give meaning to the article. The theoretical part has been completely restructured, reformulating the different headings and information provided in the first draft. The research question has been changed to really meet the method used and the procedures of this research. Additionally, an effort has been made to tone down the language, and the limitations section has been revised and the information required has been included. We have also made sure that all tables mentioned in the text are placed in the manuscript and numbered consecutively.   We hope that you will find the revised manuscript more suitable for publication. Any other questions or modifications needed we will take into account. Thank you very much for your time and effort.   Sincerely,   The Authors

Reviewer 4 Report

Accept 

Author Response

Dear revieweer,

Thank you very much for accepting the manuscript and for your helpfull coments through the proccess.

Best regards,

Author